# Patient-Derived Pancreatic Cancer Cells Induce C2C12 Myotube Atrophy by Releasing Hsp70 and Hsp90

**DOI:** 10.3390/cells11172756

**Published:** 2022-09-03

**Authors:** Hong-Yu Wu, Jose G. Trevino, Bing-Liang Fang, Andrea N. Riner, Vignesh Vudatha, Guo-Hua Zhang, Yi-Ping Li

**Affiliations:** 1Department of Integrative Biology and Pharmacology, University of Texas Health Science Center at Houston (UTHealth Houston), Houston, TX 77030, USA; 2Department of Surgery, Virginia Commonwealth University, Richmond, VA 23298, USA; 3Department of Thoracic and Cardiovascular Surgery, University of Texas MD Anderson Cancer Center, Houston, TX 77030, USA; 4Department of Surgery, University of Florida, Gainesville, FL 32611, USA

**Keywords:** pancreatic cancer, patient-derived xenografts, Hsp70, Hsp90, myotube atrophy

## Abstract

Pancreatic cancer (PC) patients are highly prone to cachexia, a lethal wasting syndrome featuring muscle wasting with an undefined etiology. Recent data indicate that certain murine cancer cells induce muscle wasting by releasing Hsp70 and Hsp90 through extracellular vesicles (EVs) to activate p38β MAPK-mediated catabolic pathways primarily through Toll-like receptor 4 (TLR4). However, whether human PC induces cachexia through releasing Hsp70 and Hsp90 is undetermined. Here, we investigated whether patient-derived PC cells induce muscle cell atrophy directly through this mechanism. We compared cancer cells isolated from patient-derived xenografts (PDX) from three PC patients who had cachexia (PCC) with those of three early-stage lung cancer patients without cachexia (LCC) and two renal cancer patients who were not prone to cachexia (RCC). We observed small increases of Hsp70 and Hsp90 released by LCC and RCC in comparison to non-cancer control cells (NCC). However, PCC released markedly higher levels of Hsp70 and Hsp90 (~ 6-fold on average) than LCC and RCC. In addition, PCC released similarly increased levels of Hsp70/90-containing EVs. In contrast to RCC and LCC, PCC-conditioned media induced a potent catabolic response in C2C12 myotubes including the activation of p38 MAPK and transcription factor C/EBPβ, upregulation of E3 ligases UBR2 and MAFbx, and increase of autophagy marker LC3-II, resulting in the loss of the myosin heavy chain (MHC ~50%) and myotube diameter (~60%). Importantly, the catabolic response was attenuated by Hsp70- and Hsp90-neutralizing antibodies in a dose-dependent manner. These data suggest that human PC cells release high levels of Hsp70 and Hsp90 that induce muscle atrophy through a direct action on muscle cells.

## 1. Introduction

Pancreatic cancer (PC) has the highest mortality rate of all cancers, with a 5-year survival rate of 11% [1]. Nearly 80% of deaths in advanced PC are associated with cachexia, a wasting syndrome that features a complex metabolic disorder with progressive weight loss, muscle atrophy, fatigue, weakness, and significant loss of appetite [2,3,4]. The frequency and severity of cachexia seen in PC patients are significantly higher than other cachexia-prone cancers including lung, gastric, and colon cancer [5]. Loss of muscle mass accounts for the bulk of the body weight loss, which is a powerful and independent predictor of poor survival in PC [6]. Ameliorating muscle wasting could significantly improve cancer survival [7]. However, an understanding of the highly complex etiology of PC-induced muscle wasting remains limited.

A primary cause of cancer-induced muscle wasting is accelerated myofibrillar protein degradation [5]. Historically, the elevation of circulating inflammatory cytokines was thought to be the major trigger of muscle mass loss in cancer [8]. IL-6 [9] and activin [10] have been shown to stimulate muscle mass loss in murine PC models. However, recent clinical data indicate that cachexia in PC patients is not necessarily associated with high levels of classical cytokines including IL-6, TNFα, and IL-1β [11]. Similar data were reported in patients with diverse gastrointestinal and genitourinary cancer [12]. On the other hand, PC and other cancer patients display elevated serum Hsp70 and Hsp90 [12,13,14,15] that are considered danger-associated molecular patterns (DAMPs) capable of inducing systemic inflammation [16]. Elevated circulating Hsp70 and Hsp90 have been shown to be responsible for muscle wasting in mice bearing Lewis lung carcinoma (LLC) or intestinal adenocarcinoma secondary to the Apc^min/+^ mutation through activating TLR4, which induces muscle wasting directly and increases circulating cytokines, including IL-6 and TNFα, to promote muscle wasting indirectly [17]. In muscle cells, the TLR4- or cytokine-mediated activation of p38β MAPK is critical to murine tumor-induced muscle wasting by stimulating protein degradation through the ubiquitin-proteasome and autophagy-lysosome pathways [18,19,20]. In patients with gastrointestinal and genitourinary cancer, p38β MAPK activity in skeletal muscle is increased, which correlates with body weight loss [12]. In addition, immortalized human pancreatic cancer cell lines AsPC-1 and BxPC-3 constitutively release high levels of Hsp70 and Hsp90 through EVs and activate p38β MAPK-mediated muscle wasting [21]. However, immortalized cancer cell lines generally recapitulate limited features of human cancer cells, and it remains to be determined whether human PC cells induce muscle wasting by releasing high levels of Hsp70 and Hsp90. Primary human cancer cells display a higher fidelity to the original cells than cell lines and are a preferred tool for identifying cancer-released cachectic factors. Unfortunately, direct cell isolation and culture from human PC specimens are of extremely low efficiency. In recent years patient-derived xenografts (PDXs) have been extensively used in translational cancer research [22]. Isolating and culturing cancer cells from PDXs are highly efficient and allow in vitro studies of cancer cells from diverse types of patients. PDX-derived PC cells closely recapitulate human PC cells [23] and key aspects of cachexia [24]. The current study aims to investigate whether PDX-derived PC cells directly induce muscle cell atrophy by releasing Hsp70 and Hsp90. We examined the effect of conditioned media of cultured human PC cells isolated from PDXs on protein catabolism in C2C12 myotubes. Our data demonstrate that human PC cells release high levels of Hsp70 and Hsp90 that induce myotube atrophy by activating p38β MAPK-mediated protein catabolism.

## 2. Materials and Methods

### 2.1. Cell Cultures

Three lines of previously established PC cells derived from PDX of patients with cachexia (PCC G46, G68 and G87) [23] were cultured in advanced Dulbecco’s Modified Eagle Medium with nutrient mixture F12, 10% fetal bovine serum (10437028, ThermoFisher Scientific, Waltham, MA, USA), 6 mmol/L glutamine (35050-061, ThermoFisher Scientific, Waltham, MA, USA), 1% of penicillin/streptomycin (SV30010, Cytiva, Marlborough, MA, USA), and 20 ng/mL EGF (PHG0311, Invitrogen, Waltham, MA, USA), at a density of 10^5^ cells/mL. For a comparison with cancer cells from patients without cachexia, two lines of previously established renal cancer cells from PDXs of renal cancer patients (RCC 11 and 15) [25] were cultured in Dulbecco’s Modified Eagle Medium supplemented with 10% fetal bovine serum. In addition, previously established lung cancer cells derived from PDXs of Stage I/II non-small cell lung cancer patients without weight loss [26,27] were cultured in RPMI 1640 medium supplemented with 10% fetal bovine serum. Non-tumorigenic human pancreatic ductal epithelial cells (HPDE, American Type Culture Collection) were cultured in Keratinocyte Serum-Free Medium supplemented with 20 ng/mL EGF, bovine pituitary extract (Invitrogen, 17005042, Waltham, MA, USA), and 1x antibiotic-antimycotic (Gibco, 15240-062, Waltham, MA, USA), and NL20 (human lung epithelial cells, American Type Culture Collection) were cultured in Dulbecco’s Modified Eagle Medium supplemented with 10% fetal bovine serum. Conditioned medium from 48-h cultures of the above cells was collected and centrifuged (1000 × g for 5 min) for the treatment of C2C12 myotubes (25% final volume in fresh medium) when indicated and replaced every 24 hrs. C2C12 myoblasts (American Type Culture Collection, ATCC) were grown in growth medium (DMEM supplemented with 10% fetal bovine serum). Myoblast differentiation was induced at 85% confluence with differentiation medium (DMEM supplemented with 4% heat-inactivated horse serum) for 96 hrs. All cultured cells were maintained at 37 °C with 95% humidity and 5% CO_2_. When indicated, cancer cell-conditioned media were treated with antibodies to Hsp70 (ADI-SPA-810) and Hsp90 (ADI-SPA-830, Enzo Life Technology, Farmingdale, NY, USA) prior to treatment for myotubes at 0.5 or 1 mg/mL. Pre-immune IgG (Millipore Sigma, Burlington, MA, USA) was used as the control for the antibody treatment. The cell culture-based experiments were replicated independently at least three times.

### 2.2. Quantification of Extracellular Hsp70 and Hsp90

The Hsp70 and Hsp90α levels in cell-conditioned medium (concentrated 20-fold by centrifugation with 10K filters from Millipore) were analyzed by enzyme-linked immunosorbent assay (ELISA) according to the manufacturer’s instructions (Enzo Life Sciences, Plymouth, PA, USA). Hsp70 and Hsp90α in the culture medium from fetal bovine serum supplement were also measured and subtracted from the total Hsp70 and Hsp90α levels measured in cell-conditioned media.

### 2.3. Extracellular Vesicle (EV) Isolation and Quantitation

Hsp70/90-containing EVs in concentrated cell-conditioned media were isolated using the ExoQuickTM kit (System Biosciences, Mountain View, Palo Alto, CA, USA) [28] and quantified by measuring the activity of acetylcholinesterase (AchE), as described previously [17]. Hsp70/90-containing EVs from fetal bovine serum supplement in the culture medium were also measured and subtracted from the total Hsp70/90-carrying EV levels measured in cell-conditioned media.

### 2.4. Western Blot Analysis

Western blot analysis was carried out as previously described [12]. Due to the limited lane number available on PAGE gel, the analyzed LCC were reduced to two lines. Antibodies to AchE (SC-11409, 1:1000), TSG101 (sc-7964, 1:1000), CD81 (sc-166029, 1:1000), and Hsp70 (SC-24, 1:1000) were from Santa Cruz Biotechnology. Antibody to Hsp90 (13171-1-AP, 2:1000) was from Proteintech. Antibody to CD63 (A5271, 1:1000) was from ABclonal (Woburn, MA, USA). Antibodies to total (9212L, 1:2000) and phosphorylated p38 MAPK (4511S, 1:1000) were from Cell Signaling Technology (Danvers, MA, USA). Antibodies for MHC (MF20, 1:5000) were from Developmental Studies Hybridoma Bank at the University of Iowa, Iowa City, IA. Antibody for Atrogin1/MAFbx (1:2000) was custom-generated by Pocono Rabbit Farm & Laboratory and verified previously (Zhang et al., 2021). Antibodies for UBR2 (NBP1-45243, 1:1000) and LC3 (NB100-2220, 1:1000) were from Novus Biologicals (Littleton, CO, USA). Antibodies for total C/EBPβ (3082S, 1:1000) and C/EBPβ phosphorylated on Thr-235 (3084, 1:1000) were from Cell Signaling Technology (Danvers, MA, USA). Antibody for β-actin (sc-47778, 1:5000) was from Santa Cruz Biotechnology (Dallas, TX, USA). The optical densities of the detected bands were normalized to loading control β-actin or Ponceau S-stained proteins, except for LC3-II, which was normalized to LC3-I.

### 2.5. Immunoprecipitation

Immunoprecipitation was conducted as previously described [29]. Briefly, conditioned medium was concentrated 10-fold and incubated with Hsp70- and Hsp90-neutralizing antibodies (ADI-SPA-810 and ADI-SPA-830, Enzo Life Technology) or mouse IgG (12-371, Sigma-Aldrich, St. Louis, MO, USA) overnight at 4 °C. The antibodies were collected using Protein A/G agarose beads (20421, ThermoScientific, Waltham, MA, USA). The resulting supernatant and pellet were analyzed by Western blotting.

### 2.6. Fluorescence Microscopy

C2C12 myotubes were stained with anti-MHC antibody (MF-20, Development Studies Hybridoma Bank at the University of Iowa, Iowa City, IA) and FITC-conjugated secondary antibody, and examined using an Olympus BX60 microscope at 40×. Photographs were taken by a camera operated with DP controller software (Olympus, Shinjuku City, Tokyo, Japan). The MHC-stained myotube diameter was measured as previously described [30]. Briefly, the diameters were measured from a total of 200 myotubes from ≥10 random fields using computerized image analysis (Scion Image, Frederick, MD, USA). The diameters were measured at three points along the myotube length to obtain the averages.

### 2.7. Statistical Analyses

Data were analyzed with a one-way analysis of variance combined with Bonferroni’s multiple comparison test using the GraphPad software, as indicated. When applicable, control samples from independent experiments were normalized to a value of 1. All data were expressed as means ± standard deviation. Statistical significance was accepted at *p* < 0.05, as indicated by * or ^#^.

## 3. Results

The conditioned media of human PC cells derived from three previously established PDX lines of patients with cachexia (PCC G46, G68, and G87) [23] were analyzed for levels of Hsp70 and Hsp90α with ELISA. For a comparison with human cancer cells that did not induce cachexia, the conditioned medium of three lines of PDX-derived lung cancer cells [31] from Stage I/II lung cancer patients who did not have cachexia (LCC 250, 429, and 464) and two lines of PDX-derived renal cancer cells that were not prone to cachexia (RCC 11 and 15) were analyzed in parallel. The conditioned medium of non-tumorigenic human pancreatic ductal endothelial cell line HPDE, which does not release an elevated level of Hsp70 and Hsp90 [17], was analyzed as a non-cancer control (NCC). We observed that NCC released minimum levels of Hsp70 and Hsp90α, while LCC and RCC released either similar or modestly increased levels of Hsp70 and Hsp90α. However, PCC released markedly higher levels of both Hsp70 and Hsp90α than RCC or LCC (average ~6-fold, Figure 1). Given that cancer cells are known to release Hsp70 and Hsp90 as membrane proteins on EVs [17,32,33], we examined whether PCC released higher levels of Hsp70- and Hsp90-containing EVs. Due to the heterogeneity of EVs and the fact that Hsp70- and Hsp90-containing EVs are associated with AchE, a non-generic marker of EVs that is associated with a subset of EVs [34], Hsp70/90-containing EVs can be quantified by measuring AchE activity in isolated EVs with a higher accuracy [17]. We verified the presence of Hsp70, Hsp90, AchE, and generic EV marker CD63, CD81, and TSG101 in EVs isolated from the conditioned medium of PCC with Western blotting (Figure 2A), and quantified Hsp70/90-containing EVs present in the conditioned media of tested cells by determining the AchE activity (Figure 2B). This revealed that NCC released essentially no Hsp70/90-containing EVs, whereas LLC and RCC released relatively low levels of such EVs. In contrast, PCC released markedly higher levels of Hsp70/90-containing EVs than LCC or RCC (~ 10-fold on average, Figure 2). These data indicate that PCC constitutively release much higher levels of Hsp70 and Hsp90 through EVs than LCC and RCC do.

To determine whether PCC-released Hsp70 and Hsp90 are responsible for activating the p38 MAPK-mediated catabolic pathway in muscle cells, C2C12 myotubes were treated with the conditioned media of PCC, LCC, and RCC to compare their catabolic activity and determine whether neutralizing Hsp70 and Hsp90 antagonizes the catabolic activity of PCC. The conditioned medium of non-tumorigenic human lung endothelial cell line NL20, which does not release an elevated level of Hsp70 and Hsp90 [17], was used as a non-cancer control (NCC). The effect of the neutralizing antibodies on the depletion of Hsp70 and Hsp90 in PCC-conditioned medium was verified by immunoprecipitating Hsp70 and Hsp90 from the medium. Figure 3A shows that Hsp70/90 were enriched in the pellet and reduced in the supernatant. The catabolic activity of PCC-conditioned medium on C2C12 myotubes was evaluated by a Western blot analysis of the catabolic pathway activated by Hsp70/90 [17], as shown in Figure 3B. PCC-conditioned medium activated p38β MAPK and its effector C/EBPβ through the phosphorylation of Thr-188 of C/EBPβ [35] in one hour, upregulated C/EBPβ-responsive E3 ligases UBR2 [36] and MAFbx [29], increased autophagy marker LC3-II, which was mediated by C/EBPβ as well as p38β MAPK [18], in eight hours, and caused the loss of MHC in 72 h. These data are very similar to previous observations from murine cancer cells [17] and immortalized human PC cells [21]. In contrast, LCC and RCC did not have these catabolic effects on myotubes. Further, neutralizing antibodies for Hsp70 and Hsp90 (0.5 and 1.0 μg/mL each) attenuated the catabolic response induced by PCC in a dose-dependent manner. To assess the PCC effect on myotube atrophy, myotubes were stained for MHC using immunofluorescence, and the myotube diameters were measured. As shown in Figure 4, LCC and RCC did not reduce the myotube diameter when compared with the conditioned medium of NCC. However, PCC reduced the myotube diameter by ~60%. Pretreatment of PCC-conditioned media with neutralizing antibodies for Hsp70 and Hsp90 attenuated the PCC-induced loss of the myotube diameter in a dose-dependent manner. These data support the notion that PCC induces myotube atrophy directly by releasing Hsp70 and Hsp90.

## 4. Discussion

The current study demonstrates that human PC cells release high levels of Hsp70 and Hsp90 to directly activate the skeletal muscle cell catabolism independently of the systemic host response to cancer. These data support the concept that extracellular Hsp70 and Hsp90 are biomarkers and therapeutic targets of PC-induced cachexia.

The elevation of circulating Hsp70 and Hsp90 has been reported in various cancer patients including cachexia-prone lung, colon, and PC patients [12,13,14,15]. In addition, circulating Hsp70 and Hsp90 in lung cancer patients correlate with the development of the pathological grade and clinical stage [15,37,38], as well as mortality [14]. Whether the increase in circulating Hsp70 and Hsp90 correlates with cachexia is yet to be established. Our findings that PCC releases dramatically higher levels of Hsp70 and Hsp90 than non-cachectic LCC and RCC to induce muscle atrophy support the concept that extracellular Hsp70 and Hsp90 are biomarkers and key mediators of PC-induced cancer cachexia. On the other hand, LCC and RCC released modestly increased levels of Hsp70 and Hsp90 over NCC without causing myotube atrophy, suggesting that these levels of Hsp70 and Hsp90 are below the threshold for inducing the catabolic response in muscle cells.

Our data demonstrate that neutralizing extracellular Hsp70 and Hsp90 attenuates PCC-induced myotube atrophy in a dose-dependent manner, supporting the notion that elevated extracellular Hsp70 and Hsp90 are key inducers of muscle catabolism by PCC. PCC-conditioned medium activated p38 MAPK and p38β MAPK effector C/EBPβ in one hour, consistent with previous findings from immortalized cancer cells [21,29,35], suggesting that this action involved Hsp70 and Hsp90 receptor TLR4-mediated post-transcriptional modifications independently of the cancer-induced systemic increase of inflammatory cytokines [17,19] that can activate p38 MAPK including IL-6 [39], TNFα [40], IL-1 [41], and activin [10,20]. In addition, the IL-6 stimulation of muscle mass loss may also involve STAT3-mediated signaling [9]. On the other hand, due to that circulating Hsp70 and Hsp90 increase with disease progression [15,37,38], in more advanced cancer patients, higher levels of circulating Hsp70 and Hsp90 may further increase circulating cytokines that exacerbate cachexia.

Extracellular Hsp70 and Hsp90 induce muscle wasting by activating TLR4 [17]. TLR4 activation increases the cellular generation of ROS [42], which stimulates cancer-induced muscle wasting [43]. TLR4-mediated muscle wasting can be ameliorated by the antioxidant agent curcumin [44]. As TLR4 effectors, the p38 MAPK family in skeletal muscle is activated by ROS and ROS-generating cytokines such as TNFα [40,45]. Thus, antioxidants may be useful in ameliorating PC-induced muscle wasting by antagonizing the Hsp70/90-induced activation of p38β MAPK, and they thus warrant future investigations.

In conclusion: the current in vitro study suggests that human PC cells release high levels of Hsp70 and Hsp90 to activate p38β MAPK-mediated muscle mass loss directly. Future in vivo and patient studies are warranted in order to determine whether circulating Hsp70 and Hsp90 are indeed biomarkers and therapeutic targets of PC-induced cancer cachexia.

## Figures and Tables

**Figure 1 cells-11-02756-f001:**
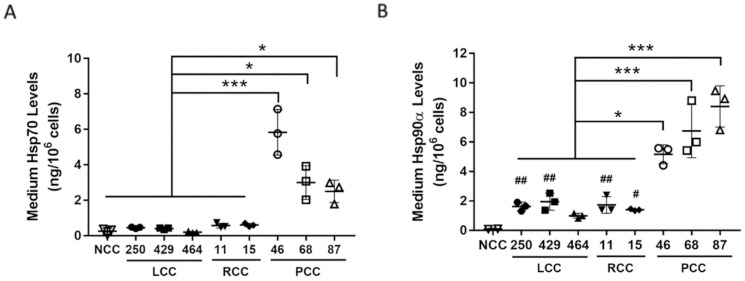
PCC releases markedly higher levels of Hsp70 and Hsp90 than LCC and RCC. Forty-eight-hour culture media of NCC, LCC, RCC, and PCC cells were analyzed for content of (**A**) Hsp70 and (**B**) Hsp90a using ELISA. Data (means ± SD) from 3 independent experiments were analyzed by one-way ANOVA followed by Bonferroni’s multiple comparison test. ^#^ denotes difference from NCC, and * denotes difference from LCC and RCC at *p* < 0.05. ^##^ denotes *p* < 0.01. *** denotes *p* < 0.001.

**Figure 2 cells-11-02756-f002:**
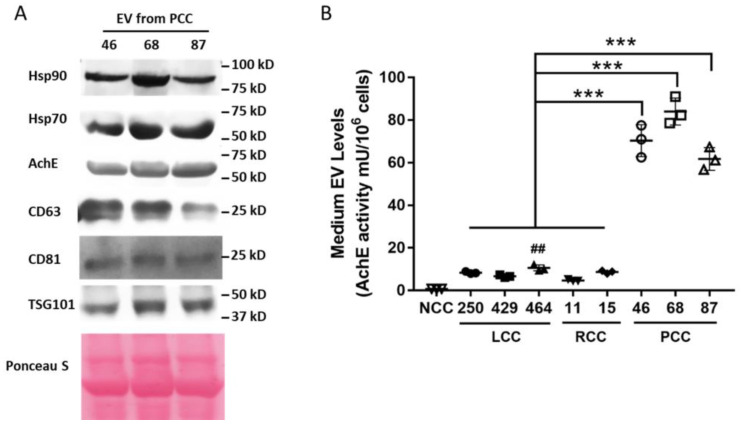
PCC releases markedly higher levels of Hsp70- and Hsp90-carrying EVs than LLC and RCC. (**A**) EVs isolated from PCC-conditioned medium were analyzed for Hsp70/90 and protein markers with Western blotting. (**B**) Hsp70/90-containing EVs were quantified by AchE activity in isolated EVs. Data from 3 independent experiments were analyzed by one-way ANOVA followed by Bonferroni’s multiple comparison test. ^##^ denotes difference from NCC at *p* < 0.01, and *** denotes difference from LCC and RCC at *p* < 0.001.

**Figure 3 cells-11-02756-f003:**
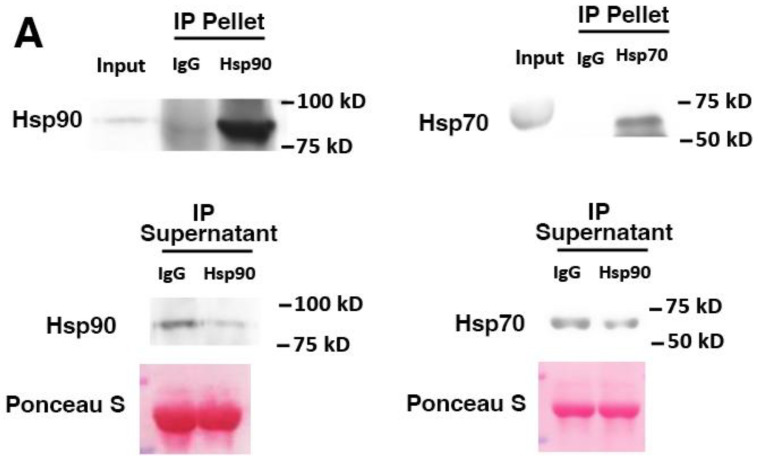
PCC activates p38β MAPK-mediated catabolic pathways in myotubes by releasing Hsp70 and Hsp90. C2C12 myotubes were treated with conditioned medium of PPC cells (68, 87, and 46), LCC cells (464 and 250), and RCC cells (11 and 15) for a comparison with NCC. The media were supplemented with either neutralizing antibodies against Hsp70 and Hsp90 (0.5 and 1.0 μg/mL each) or pre-immune IgG (control), as indicated. (**A**) Binding of neutralizing antibodies to Hsp70 and Hsp90 present in conditioned medium was verified by immunoprecipitation. Hsp70 and Hsp90 contents in resulting pellet and supernatant were analyzed by Western blotting. (**B**) Catabolic activity of cancer cell-conditioned media on C2C12 myotubes was monitored by Western blot analysis of the activation of p38 MAPK and C/EBPβ by specific site phosphorylation at 1 h, levels of E3 ligases (UBR2 and MAFbx) and LC3-II at 8 h, and levels of MHC at 72 h. Three independent experiments (*n* = 3) were carried out. Data were analyzed by one-way ANOVA followed by Bonferroni’s multiple comparison test. ^#^ denotes difference from NCC, and * denotes difference from LCC and RCC at *p* < 0.05. ^##^ denotes difference from NCC at *p* < 0.01.

**Figure 4 cells-11-02756-f004:**
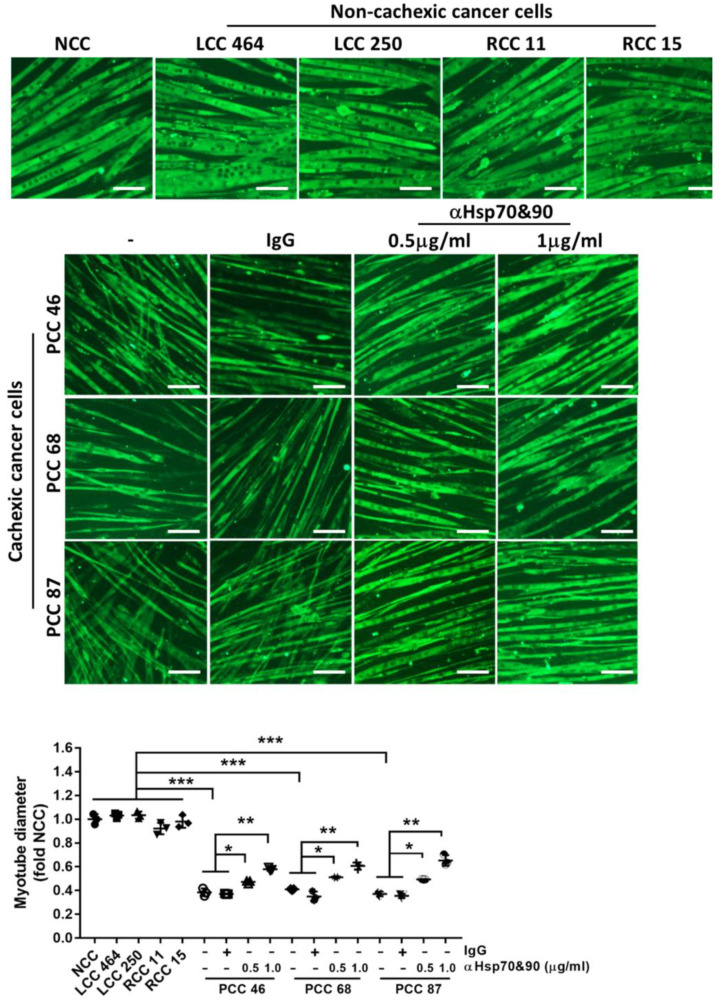
Conditioned medium of PCC causes myotube atrophy in an Hsp70- and Hsp90-dependent manner. C2C12 myotubes were treated as described in Figure 3 for 72 h. Myotubes were subjected to immunofluorescence staining of MHC. The myotube diameter was measured. Bar = 100 μm. Three independent experiments (*n* = 3) were carried out. Data were analyzed by one-way ANOVA followed by Bonferroni’s multiple comparison test. * denotes *p* < 0.05, ** denotes *p* < 0.01 and *** denotes *p* < 0.001.

## Data Availability

No data were reported by this study.

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
