# Peer review of "Patient-Derived Pancreatic Cancer Cells Induce C2C12 Myotube Atrophy by Releasing Hsp70 and Hsp90"

_cells, 2022, doi:10.3390/cells11172756_

Round 1

Reviewer 1 Report

This is a small report where authors have investigated the role of HSP70 and HSP90 proteins released by Patient-Derived Pancreatic Cancer Cells (PCC) on C2C12 myotube. They report that compared to lung or renal cancer cells, PCC release higher amounts of HSP70 and HSP90 in conditioned medium. Authors demonstrate that PCC conditioned medium activates p38MAPK, C/EBP, MAFbx E3 ubiquitin ligase, and LC3B marker of autophagy, which are inhibited by neutralizing antibodies for HSP70 and HSP90. The lead author on this manuscript has published several articles demonstrating similar mechanisms of cancer cachexia in animal models. The present study demonstrate that human patients derived pancreatic tumor cells also induce cachexia through similar mechanisms. The manuscript is well written and the experiments are easy to follow. I have a few minor suggestions.

1)  To improve the clinical relevance of the findings in the manuscript, authors should try to perform some of these experiments on human primary myotubes, which are commercially available and widely used.

2) Authors should consider some additional analysis of markers of UPS and autophagy. There is also a concern whether the MAFbx protein bands shown in Fig 3 is real. The uncropped blot included with the manuscript has multiple bands and all the bands show similar increase or decrease. MAFbx antibody is not so good. Therefore, it is important to validate these findings by QRT-PCR. Similar concern is for p-p38 immunoblots. Multiple bands and all are showing similar changes. Can author measure some downstream targets of p38 MAPK by Western blot or QRT-PCR?

3) Cancer cachexia involves multiple signaling pathways. Authors should investigate or comment whether conditioned medium of PCC affects the activation of other pathways and mechanisms.

Author Response

Reviewer 1

This is a small report where authors have investigated the role of HSP70 and HSP90 proteins released by Patient-Derived Pancreatic Cancer Cells (PCC) on C2C12 myotube. They report that compared to lung or renal cancer cells, PCC release higher amounts of HSP70 and HSP90 in conditioned medium. Authors demonstrate that PCC conditioned medium activates p38MAPK, C/EBP, MAFbx E3 ubiquitin ligase, and LC3B marker of autophagy, which are inhibited by neutralizing antibodies for HSP70 and HSP90. The lead author on this manuscript has published several articles demonstrating similar mechanisms of cancer cachexia in animal models. The present study demonstrate that human patients derived pancreatic tumor cells also induce cachexia through similar mechanisms. The manuscript is well written and the experiments are easy to follow. I have a few minor suggestions.

  1. Thank you for your precious time reviewing this manuscript and insightful comments.

1)  To improve the clinical relevance of the findings in the manuscript, authors should try to perform some of these experiments on human primary myotubes, which are commercially available and widely used.

  1. We have thought about this ourselves, too. However, technical difficulties prevented us from doing it. The main difficulty is that human myotubes contract strongly to the extent that they pull themselves off the plate about 2 days after completing differentiation.  Our experiments lasted for 72 hours after differentiation, which is the period necessary for observing loss of MHC and myotube atrophy.  On the other hand, C2C12 myotubes do not pull themselves off the plate.  Importantly, C2C12 myotubes recapitulate many aspects of cancer-induced muscle atrophy in vivo.  Due to these issues, C2C12 myotubes, not human myotubes, have been used extensively in cancer cachexia studies. 

2) Authors should consider some additional analysis of markers of UPS and autophagy. There is also a concern whether the MAFbx protein bands shown in Fig 3 is real. The uncropped blot included with the manuscript has multiple bands and all the bands show similar increase or decrease. MAFbx antibody is not so good. Therefore, it is important to validate these findings by QRT-PCR. Similar concern is for p-p38 immunoblots. Multiple bands and all are showing similar changes. Can author measure some downstream targets of p38 MAPK by Western blot or QRT-PCR?

  1. Our previous publications demonstrated that cancer activates the UPP and the ALP in skeletal muscle.Due to the focus of this study on Hsp70/90-mediated signaling in muscle cells, we selected markers of UPS and autophagy that are known to be regulated by p38b MAPK in response to Hsp70/90 including transcription factor C/EBPb, E3 MAFbx and UBR2, and autophagy marker LC3-II.  We have enhanced the background information on why we selected these markers by adding relevant references for each of the molecules monitored.  

The custom antibody for MAFbx was validated previously using siRNA knockdown of MAFbx in the Supplement Information of our paper - Zhang et al., Front. Cell Dev. Biol., 9:784424, 2021, which is shown below in Figure 1.  As a polyclonal antibody it detected more than one band, however, the 42 kDa band of MAFbx is the prominent band (indicated by the arrow on uncropped blot).  We have shown previously that changes in MAFbx level detected by this antibody were consistent to changes in its mRNA in muscle of cancer patients (Zhang et al., Front. Cell Dev. Biol., 9:784424, 2021), and the data are copied in Figure 2.

Figure 1.  Validation of the specificity of custom antibody against Atrogin1/MAFbx.  C2C12 myoblasts were transfected with control or Atrogin1/MAFbx-specific siRNA (SASI_Mm01_00036617, Sigma-Aldrich) using the jetPRIME reagent (Polyplus-transfection Inc., Illkirch, France).  After differentiation for 96 h, myotubes were lysed and analyzed for Atrogin1/MAFbx protein level with Western blotting using a custom antibody generated by Pocono Rabbit Farm & Laboratory (Pocono, PA) using a peptide (NILEKVVLKVLE + C and KLH conjugation) synthesized by Bon Opus Biosciences (Millburn, JJ).

Figure 2.  Higher expression of p38b MAPK ® C/EBPb-regulated genes in UPP and ALP in rectus abdominis A of cancer patients.  (A) The protein levels of E3s implicated in muscle wasting were analyzed by Western blotting.  (B) The mRNA levels of C/EBPb-responsive genes in the UPP and ALP were analyzed by qPCR.  The mRNA of TLR4 was monitored as a negative control.   Data were analyzed by one-way ANOVA combined with Tukey’s test.  * denotes p < 0.05 compared to control.

3) Cancer cachexia involves multiple signaling pathways. Authors should investigate or comment whether conditioned medium of PCC affects the activation of other pathways and mechanisms.

  1. Certainly, cancer cachexia involves multiple signaling pathways.This study focuses on pancreatic cancer-induced cachexia, which may have unique features.  Dr. Teresa Zimmers’ group recently showed that murine pancreatic cancer cells (KPC) release IL-6 to stimulate muscle wasting, which may involve signaling through STAT3 (Rupert et al. J Exp Med (2021) 218 (6): e20190450).  We have included this info in discussion (page 14). 

Reviewer 2 Report

In the current manuscript, the authors demonstrated that human pancreatic cancer (PC) cells release high levels of Hsp70 and Hsp90 through EVs to directly activate skeletal muscle cell catabolism by activation of p38 MAPK pathways. The results are clearly indicated and support the concept that extracellular Hsp70 and Hsp90 could be biomarkers of PC-induced cachexia. However, the molecular basis of the contribution of Hsps to myotube atrophy is not clarified.  

Major comments:

1. In Figure1, Hsp70 and Hsp90alpha were quantified by ELISA and high levels of Hsps were detected from PCC. And the high level of EV marker AchE was also detected from the PCC-conditioned medium in Figure2. How could the authors conclude that Hsps were the contents of EV in these experiments? The HSPs could be secreted or come from the dead cells in the culture. Please justify it.

2.  Figure3 is too small to read.

3. Figure 4 showed that antibodies to Hsps improved the atrophy of C2C12 myotubes. From our experience, it is hard to believe that the myotube diameter could be measured with such small variances. A more detailed description of the myotube diameter measurement is needed in the materials and methods section, and all the measured images should be presented for reviewers.   

4. Addition of antibodies to Hsps affected many factors in figure3, but they did not show immunoblotting of Hsp70 and Hsp90. It is very important to show the data to confirm that the addition of antibodies really depleted the target proteins.

5. If Hsp70 and Hsp90 are the major factors that cause muscle atrophy, the addition of recombinant Hsps could cause atrophy? The authors should mention how Hsps could activate TLR4 in the manuscript.

Author Response

Review 2

In the current manuscript, the authors demonstrated that human pancreatic cancer (PC) cells release high levels of Hsp70 and Hsp90 through EVs to directly activate skeletal muscle cell catabolism by activation of p38 MAPK pathways. The results are clearly indicated and support the concept that extracellular Hsp70 and Hsp90 could be biomarkers of PC-induced cachexia. However, the molecular basis of the contribution of Hsps to myotube atrophy is not clarified.  

  1. Thank you for your precious time reviewing this manuscript and insightful comments.

Major comments:

  1. In Figure1, Hsp70 and Hsp90alpha were quantified by ELISA and high levels of Hsps were detected from PCC. And the high level of EV marker AchE was also detected from the PCC-conditioned medium in Figure2. How could the authors conclude that Hsps were the contents of EV in these experiments? The HSPs could be secreted or come from the dead cells in the culture. Please justify it.
  2. We have previously established that cancer cells release EVs containing Hsp70 and Hsp90 that induce muscle wasting (Zhang et al., Nat Commun 8, 589. 2017). We copied some figures from that paper to demonstrate this point. The morphology (panel A) and size (panel A and C) of our EV preparations from LLC cell conditioned medium and serum of LLC tumor-bearing mice are consistent to those of exosomes.  The protein content of the EV preparations from cachectic cancer cells were all elevated (panel B), yet in a somewhat different proportion compared to the elevation of AchE activity in EVs (panel E). The EVs contained CD9/TSG101/AchE along with Hsp70 and Hsp90, and immunoprecipitation of CD9-positive EVs confirmed the association of AchE with Hsp70/90-positive EVs but not supernatant (panel D).  These data demonstrated that AchE is associated with a subset of EVs that contained Hsp70 and Hsp90.  Thus it provides a more accurate measurement of Hsp70/90-containing EVs than protein content of isolated EVs.

Subsequently we observed similar results in immortalized human pancreatic cancer cell line AsPC-1 (Yang et al., Gastroenterology156, 722-734 e726. 2019).  We copied Figure 6C from that paper demonstrating association of Hsp70/90 with AchE-positive EVs released by AsPC-1 cells, and dependence of the EV release on ZIP4. ®

The current study showed that PDX-derived pancreatic cancer cells similarly release high levels of Hsp70 and Hsp90 as well as AchE-positive EVs.  To address your concern, we now have added Western blotting data confirming that EVs isolated from PCC conditioned medium contained Hsp70, Hsp90, AchE and generic EV markers (Figure 2A).  The key point of this study is that PDX-derived pancreatic cancer cells induce muscle atrophy through the release of Hsp70 and Hsp90.  EVs are released by viable cells.  The high level release of AchE-positive EVs by PCC indicates the health of the cells.  We did not use chemotherapeutic drugs or other toxic substrances in the experiment, thus, there is minimum cell death if any. 

  1. Figure3 is too small to read.
  2. The figure has been rearranged to make it larger.
  3. Figure 4 showed that antibodies to Hsps improved the atrophy of C2C12 myotubes. From our experience, it is hard to believe that the myotube diameter could be measured with such small variances. A more detailed description of the myotube diameter measurement is needed in the materials and methods section, and all the measured images should be presented for reviewers.  
  4. The method of measuring myotube diameters were previously adopted from Menconi et al. J. Cell. Biochem. 105, 353–364, 2008. We have routinely used this method and published multiple papers including Doyle et al. FASEB J. 25, 99–110, 2011; Zhang et al., Nat. Commun. 19;8(1):589, 2017; Yang et al., Gastroenterology 156:722-734, 2019; Sin et al. Cancer Res. 81(4):885-897, 2021; Liu et al. J Cachexia Sarcopenia Muscle 13(1):636-647, 2022. We have added more details of this method to the revised manuscript (page 8). Requested original photos are shown in a separate file. 
  5. Addition of antibodies to Hsps affected many factors in figure3, but they did not show immunoblotting of Hsp70 and Hsp90. It is very important to show the data to confirm that the addition of antibodies really depleted the target proteins.
  6. Now we have added new data that neutralizing antibodies depleted Hsp70 and Hsp90 in PCC conditioned medium (Figure 3A).
  7. If Hsp70 and Hsp90 are the major factors that cause muscle atrophy, the addition of recombinant Hsps could cause atrophy? The authors should mention how Hsps could activate TLR4 in the manuscript.
  8. Yes, we have shown previously that addition of recombinant Hsp70/90 causes muscle wasting in vitro and in vivo (Zhang et al., Nat Commun 8, 589. 2017). Figure 4 of that paper demonstrating this point is copied below. Circulating Hsp70/90 are known to act as DAMPs and activate TLR4 (Tamura et al. Curr. Mol. Med. 12, 1198–1206, 2012).  We showed previously that TLR4 in muscle cells is required for Hsp70/90-induced muscle wasting (Zhang et al., Nat Commun 8, 589. 2017).  We have provided related background information in the text.

Reviewer 3 Report

The manuscript by Wu et. Al., entitled, “Patient-Derived Pancreatic Cancer Cells Induce C2C12 Myo-2 tube Atrophy by Releasing Hsp70 and Hsp90” is though nicely composed yet has many gaps that need to be filled for eventual publication in the journal. Following comments may help the authors in further improving the overall concerns of the readers.

1. Weight loss and muscle atrophy is something that is common feature of all cancers? What made the group to ponder that this phenomenon must be limiting or well exhibited in pancreatic cancer itself at first place and also only also is aggressiveness of cancer the main culprit of HSP-70 & HSP-90 release and is it something that can be co-related to grade of cancer? Can authors provide a solid ground for not using any other cancer that too at the advanced stage as in the manuscript it has been demarcated benign lung cancer was used and not advanced stage and least aggressive renal cancer? The experiment would have been more meaningful with usage of equally aggressive breast, lung cancer to deduce the conclusion that cachexia is prevalent in pancreatic cancer?

2. Authors have demarcated that muscle wasting is initially induced by interleukins and then demarcated that it is not associated with higher levels of interleukins in some respectable pancreatic tumors? Term respectable needs proper explanation?? Does group want to convey that initially job for muscle atrophy is being done by interleukins and then when cancer has substantially reached higher grades of malignancy HSP’s takes over the job.  Can cachexia be assumed a case of cross talk between external tumor micro-environment and internal cellular circuitry.

3. Dexamethasone has been added in media and that too only for pancreatic carcinoma cells. I believe that group would be well versed with the fact that dexamethasone is a carcinogen or cell proliferative factor. Is its addition a sort of deliberate attempt to induce cachexia by pancreatic cells by faking higher degree of malignancy? In light of this it is suggested to redo the experiment without its addition or come up with a strong evidence that it is not contributing to the above mentioned factor.

4. Rather than using the conditioned media other way out was to opt for co-culturing both the cell types involved? What is the explanation for not opting for the same.

5. Co-relation between acetylcholinesterase activity and EV containing HSP70/90’s need to be explained a bit and incorporation of same would be highly appreciated.

6. Though all the antibodies have been mentioned but the used dilutions are missing. Authors need to mention the used dilutions for mentioned antibodies.

7. In figure 1 & 2 while the labelling of Y axis is well demarcated labelling of numeric values on the X axes is not clear. Kindly specify.

8. What does the term MHC in manuscript refer to?

9. Molecular weight for actin is 42kda while as 37kda is mentioned in the manuscript. Needs to be Rectified.

10. It is appreciable to see incorporation of immunofluorescence experiment which clearly indicates muscle atrophy upon usage of PCC conditioned media. However, proper cause of atrophy may be/is due to apoptosis or cell death, autophagy, necrosis. Proper assays needs to be carried out to ascertain it.

11. There is not a single experiment in the manuscript that demonstrates EV containing HSP’s bind solely toTLR-4 and blockade of TLR-4 can avert the pathogenesis.

12. In the discussion section, While the authors mention it is the ROS mediated cell death/apoptosis that is the main culprit of cachexia. But at the same time many cancer researchers believe that resuming ROS mediated cell death via chemotherapy or radiotherapy in cancers can improve the prognosis in cancer patients. The paradox needs to be explained.

Author Response

Review 3

The manuscript by Wu et. Al., entitled, “Patient-Derived Pancreatic Cancer Cells Induce C2C12 Myo-2 tube Atrophy by Releasing Hsp70 and Hsp90” is though nicely composed yet has many gaps that need to be filled for eventual publication in the journal. Following comments may help the authors in further improving the overall concerns of the readers.

  1. Thank you for your precious time for reviewing this manuscript and insightful comments.
  2. Weight loss and muscle atrophy is something that is common feature of all cancers? What made the group to ponder that this phenomenon must be limiting or well exhibited in pancreatic cancer itself at first place and also only also is aggressiveness of cancer the main culprit of HSP-70 & HSP-90 release and is it something that can be co-related to grade of cancer? Can authors provide a solid ground for not using any other cancer that too at the advanced stage as in the manuscript it has been demarcated benign lung cancer was used and not advanced stage and least aggressive renal cancer? The experiment would have been more meaningful with usage of equally aggressive breast, lung cancer to deduce the conclusion that cachexia is prevalent in pancreatic cancer?
  3. Weight loss and muscle atrophy is not a common feature of all cancers. It is known that pancreatic cancer has the highest prevalence and severity of cachexia, followed by gastro-oesophageal cancer, head and neck cancers and lung cancer. On the other hand, breast cancer has much lower prevalence and severity of cachexia (Baracos et al., Nat Rev Dis Primers 4, 17105).  This is why we studied PCC that represents the most cachectic cancer.  Cachexia is usually developed in the later stages of cancer.  To compare Hsp70 and Hsp90 release in later stage cancer cells that cause cachexia with earlier stage cancer cells that do not induce cachexia, we used PDX cells derived from early stage lung cancer patient who did not have cachexia (their cancer was malignant).  On the other hand, RCC represented cancers that have lowest probability of developing cachexia.  Of course, comparing more cancer cell types would be nice, but we were limited by the availability of PDX cells. 
  4. Authors have demarcated that muscle wasting is initially induced by interleukins and then demarcated that it is not associated with higher levels of interleukins in some respectable pancreatic tumors? Term respectable needs proper explanation?? Does group want to convey that initially job for muscle atrophy is being done by interleukins and then when cancer has substantially reached higher grades of malignancy HSP’s takes over the job.  Can cachexia be assumed a case of cross talk between external tumor micro-environment and internal cellular circuitry. 
  5. We apologize that we did not make the related statements very clear. We have made revisions in introduction by indicating that: Historically, elevation of circulating inflammatory cytokines was thought the major trigger of muscle mass loss in cancer (Fearon et al., 2012). IL-6 (Rupert et al., 2021) and activin (Zhong et al., 2019)have been shown to stimulate muscle mass loss In murine PC models.  However, recent clinical data indicate that cachexia in PC patients is not necessarily associated with high levels of classical cytokines including IL-6, TNFa and IL-1b (Talbert et al., 2018).  Similar data were reported in patients with diverse gastrointestinal and genitourinary cancer (Zhang et al., 2021).  On the other hand, PC and other cancer patients display elevated serum Hsp70 and Hsp90 (Dutta et al., 2012; Shi et al., 2014; Suzuki et al., 2006; Zhang et al., 2021) that are considered danger-associated molecular patterns (DAMPs) capable of inducing systemic inflammation (Bianchi, 2007) (page 3).   We have also added the following in discussion: On the other hand, due to that circulating Hsp70 and Hsp90 increase with disease progression (Balazs et al., 2016; Rong et al., 2014; Shi et al., 2014), in more advanced PC patients higher levels of circulating Hsp70 and Hsp90 may further increase circulating cytokines that exacerbate cachexia (page 14).  We hope that these revisions convey our view clearly.
  6. Dexamethasone has been added in media and that too only for pancreatic carcinoma cells. I believe that group would be well versed with the fact that dexamethasone is a carcinogen or cell proliferative factor. Is its addition a sort of deliberate attempt to induce cachexia by pancreatic cells by faking higher degree of malignancy? In light of this it is suggested to redo the experiment without its addition or come up with a strong evidence that it is not contributing to the above mentioned factor. 
  7. Thank you very much for catching this oversight. Our culture medium for PCC actually did not contain dexamethasone, instead it contained 20 ng/ml EGF. We are very sorry for this mistake and have corrected it in the revised manuscript.  We are fully aware of the catabolic effect of dexamethasone on muscle mass and would not have used it in culture medium. 
  8. Rather than using the conditioned media other way out was to opt for co-culturing both the cell types involved? What is the explanation for not opting for the same.
  9. Cancer does not induce muscle wasting by directly contacting muscle cells. Cancer cell conditioned medium simulates the cancer environment to which muscle cells are exposed to better than co-cultures. In addition, myotubes are maintained in medium containing 2 to 4% horse serum while cancer cells require the support of 10% FBS at least. The different requirements for serum supplement make it difficult to co-culture myotubes with cancer cells.  We treated myotubes with 48 hour-culture of cancer cell conditioned media (25% final volume), in which FBS has been largely consumed.  We have published multiple manuscripts using this approach (Zhang et al.,  EMBO J, 30:4323-35, 2011; Zhang et al., Nat. Commun. 19;8(1):589, 2017; Yang et al., Gastroenterology  156:722-734, 2019; Sin et al., Cancer Res. 79(7):1331-1342, 2019 and Sin et al., Cancer Res. 81:885–97, 2021).  Most of published in vitro work on cancer cachexia used this approach too. 
  10. Co-relation between acetylcholinesterase activity and EV containing HSP70/90’s need to be explained a bit and incorporation of same would be highly appreciated.
  11. Please see our response to Comment 1 by Reviewer 1. New data has been added in Figure 2A.
  12. Though all the antibodies have been mentioned but the used dilutions are missing. Authors need to mention the used dilutions for mentioned antibodies.
  13. Dilution information has been added. 
  14. In figure 1 & 2 while the labelling of Y axis is well demarcated labelling of numeric values on the X axes is not clear. Kindly specify.
  15. Figure 1 and 2 have been revised as requested.
  16. What does the term MHC in manuscript refer to?
  17. MHC is abbreviation for “myosin heavy chain”, which has been added to the text.
  18. Molecular weight for actin is 42kda while as 37kda is mentioned in the manuscript. Needs to be Rectified.
  19. We have now added the 50 kDa marker in the figure. Actin is located between 37 and 50 kDa (42 kDa).
  20. It is appreciable to see incorporation of immunofluorescence experiment which clearly indicates muscle atrophy upon usage of PCC conditioned media. However, proper cause of atrophy may be/is due to apoptosis or cell death, autophagy, necrosis. Proper assays needs to be carried out to ascertain it.
  21. Although the mechanism of cancer-induced muscle wasting is highly complex, there is a general consensus that accelerated protein degradation is a major contributor. Particularly, loss of myofibrillar proteins that constitute the bulk of muscle proteins is thought mediated by the ubiquitin proteasome system. This study is an extension of our previous publications demonstrating that cancer cell released Hsp70/90 stimulate MHC loss mediated by the ubiquitin proteasome pathway.  In addition, this signalling pathway activates autophagy.  Therefore, we monitored E3s including MAFbx and UBR2 and autophagy marker LC3-II, as well as MHC.  Our data support our hypothesis. On the other hand, we don’t exclude other potential contributors to the wasting process. 
  22. There is not a single experiment in the manuscript that demonstrates EV containing HSP’s bind solely toTLR-4 and blockade of TLR-4 can avert the pathogenesis.
  23. We have shown previously a requirement of TLR-4 for Hsp70/90 to induced muscle wasting in vitro and in vivo through activating p38b MAPK-mediated muscle catabolism (Zhang et al., Nat. Commun. 19;8(1):589, 2017). Figure 8 of that paper is copied here to demonstrate the in vitro effect.  For in vivo data please see Figure 9 of the paper.  In the current study we verified that PPC induces myotube atrophy through releasing Hsp70/90 that activate p38b MAPK-mediated muscle catabolism.  We have enhanced the explanation of this point.
  24. In the discussion section, While the authors mention it is the ROS mediated cell death/apoptosis that is the main culprit of cachexia. But at the same time many cancer researchers believe that resuming ROS mediated cell death via chemotherapy or radiotherapy in cancers can improve the prognosis in cancer patients. The paradox needs to be explained.
  25. ROS is a double edged sword. For example, TNFa got its name from its activity of causing tumor cell necrosis. It also stimulates muscle wasting.  TNFa stimulates ROS generation in muscle cells as well as in cancer cells.  Although ROS have a physiological role in cell signalling, inflammation stimulated ROS production is harmful to muscle.  Indeed, chemotherapy and radiotherapy also promote muscle wasting while injuring/killing cancer cells.  On the other hand, chemotherapy agents have diverse mechanisms in addition to simulating ROS production. 

Round 2

Reviewer 2 Report

Most of my comments were answered and I think that the ms is suitable for publication in Cells after Fig3 is adjusted properly. 

Reviewer 3 Report

Revisions done to satisfaction